# Rotavirus A during the COVID-19 Pandemic in Brazil, 2020–2022: Emergence of G6P[8] Genotype

**DOI:** 10.3390/v15081619

**Published:** 2023-07-25

**Authors:** Meylin Bautista Gutierrez, Rosane Maria Santos de Assis, Juliana da Silva Ribeiro de Andrade, Alexandre Madi Fialho, Tulio Machado Fumian

**Affiliations:** Laboratório de Virologia Comparada e Ambiental, Instituto Oswaldo Cruz, Fundação Oswaldo Cruz, Rio de Janeiro 21040-900, Brazil; meylinbautistag@gmail.com (M.B.G.);

**Keywords:** Rotavirus A, acute gastroenteritis, epidemiological surveillance, genotyping, unusual genotype, COVID-19 pandemic, Brazil

## Abstract

Rotavirus A (RVA) remains a leading cause of acute gastroenteritis (AGE) hospitalizations in children worldwide. During the COVID-19 pandemic, a reduction in vaccination coverage in Brazil and elsewhere was observed, and some reports have demonstrated a reduction in AGE notifications during the pandemic. This study aims to investigate the diversity and prevalence of RVA genotypes in children and adults presenting with AGE symptoms in Brazil during the COVID-19 pandemic between 2020 and 2022. RVA was screened using RT-qPCR; then, G and P genotypes were characterized using one-step multiplex RT-PCR. A total of 2173 samples were investigated over the three-year period, and we detected RVA in 7.7% of samples (n = 167), being 15.5% in 2020, 0.5% in 2021, and 13.8% in 2022. Higher RVA prevalence was observed in the Northeastern region (19.3%) compared to the Southeastern (6.1%) and Southern regions (5.5%). The most affected age group was children aged between 0 and 6 months old; however, this was not statistically significant. Genotyping and phylogenetic analysis identified the emergence of G6P[8] during the period; moreover, it was detected in 10.6% of samples in 2020 and in 83.5% in 2022. In contrast, the prevalence of G3P[8], the previous dominant genotype, decreased from 72.3% in 2020 to 11.3% in 2022. We also identified unusual strains, such as G3P[9] and G9P[4], being sporadically detected during the period. This is the first report on the molecular epidemiology and surveillance of RVA during the COVID-19 pandemic period in Brazil. Our study provides evidence for the importance of maintaining high and sustainable levels of vaccine coverage to protect against RVA disease. Furthermore, it highlights the need to maintain nationwide surveillance in order to monitor future trends and changes in the epidemiology of RVA in Brazil.

## 1. Introduction

Acute gastroenteritis (AGE) remains a prevailing and important cause of disease and death among children under the age of five, especially in low- and middle-income countries [1]. Group A rotavirus (RVA) is the leading cause of viral acute gastroenteritis (AGE) in children aged <5 years, exhibiting an estimated annual global mortality of 128,500 individuals, in addition to millions of hospitalizations, despite an increasing rollout of rotavirus vaccines across several countries worldwide [2,3].

RVA is a member of the family *Sedoreoviridae*, genus *Rotavirus* (https://ictv.global/report/chapter/sedoreoviridae/sedoreoviridae, accessed on 28 June 2023), and possesses a triple-layered capsid, enclosing eleven segments of double-stranded RNA (dsRNA) which encode six viral structural proteins (VP1–VP4, VP6, and VP7) and six non-structural ones (NSP1–NSP6) [4]. The outer capsid proteins, VP7 (capsid glycoprotein) and VP4 (spike protein), independently elicit neutralizing antibodies and form the basis of the binary classification system of G and P types, respectively [5]. In addition to the binary classification, the expanded RVA nomenclature includes other structural and non-structural genes segments, the identities of which are described using genome-wide genotype nomenclature, such as Gx-P[x]-Ix-Rx-Cx-Mx-Ax-Nx-Tx-Ex-Hx, with x indicating the number of corresponding segment genotypes [6].

Currently, 42 G and 58 P genotypes are recognized to infect humans and animals (https://rega.kuleuven.be/cev/viralmetagenomics/virus-classification/rcwg, accessed on 28 June 2023). Clinically, genotypes G1P[8], G2P[4], G3P[8], G4P[8], G9P[8], and G12P[8] are the most common RVA strains associated with AGE in children under five years old globally [7]. Additionally, uncommon genotypes, such as G1P[6], G2P[6], G3P[6], G3P[9], G8P[4], G9P[4], and G9P[6], have been detected and are usually related to interspecies transmission events that increase RVA genetic diversity [8,9].

Four World Health Organization (WHO)-prequalified, live-attenuated oral RVA vaccines are internationally available: Rotarix™, RotaTeq™, Rotavac™, and RotaSiil™ [10]. Moreover, as of January 2023, 123 countries have introduced RVA vaccines in their national immunization programs (NIP) [VIEW-hub by IVAC/platform on vaccine use and impact] (https://www.jhsph.edu/ivac/wp-content/uploads/2018/04/VIEW-hub_Report_Jan2023.pdf, accessed on 29 June 2023). Despite the extensive use of RVA vaccines globally, RVA infection remains a major cause of diarrhea-associated disease and death in young children [3,11]. RVA vaccines are highly effective at reducing the burden of RVA-associated AGE (deaths, hospitalizations, and severe disease) in high-income settings; however, lower effectiveness has been reported in low- and middle-income countries [12,13]. Brazil implemented the Rotarix™ vaccine into its NIP in March 2006, which led to a significant reduction in diarrhea-associated mortality and hospitalization [14].

During the COVID-19 pandemic, a reduction in vaccination coverage concerning children and adolescents was observed [15,16]. The health measures adopted because of the pandemic affected the vaccination actions that required the presence of individuals to administer the health services [17,18]. In Brazil, RVA vaccination coverage decreased from 85.4% in 2019 to 77.3% in 2020 and 70.4% in 2021. The decline was more pronounced in the second year of the pandemic, despite the fact that the circulation restrictions were already less tight [19,20].

Nationwide surveillance studies are crucial to monitor future trends and changes in the epidemiology of RVA, especially considering the downward trend in RVA vaccine coverage observed in Brazil. The present study describes RVA prevalence, epidemiology, and genotypes’ circulation in Brazil during the COVID-19 pandemic. Stool samples from medically attended AGE (outpatients and inpatients) cases were tested from three Brazilian regions (Southern, Southeastern, and Northeastern) between January 2020 and December 2022. This is the first report on the epidemiological features of RVA infection in Brazil during the COVID-19 pandemic, and we demonstrated an emergence of the G6P[8] genotype, replacing the previous dominant equine-like G3P[8] genotype.

## 2. Materials and Methods

### 2.1. Study Population and Stool Collection

This study included stool samples from medically attended AGE patients collected between January 2020 and December 2022. AGE was defined as the sudden onset of diarrhea (≥three liquid/semi-liquid evacuations within 24 h) that may be accompanied by fever, nausea, vomiting, or abdominal pain.

Stool samples were systematically sent to the Regional Rotavirus Reference Laboratory (RRRL) through sentinel sites at States Central Laboratories. The RRRL is part of the national rotavirus surveillance program, overseen by the General Coordination of Public Health Laboratories within the Brazilian Ministry of Health (MoH).

### 2.2. Ethics Statements

This study was approved by the Oswaldo Cruz Foundation (Fiocruz) Ethics Committee (approval number CAAE: 94144918.3.0000.5248). The surveillance was performed through a hierarchical network in which samples are provided by medical requests in hospitals and health centers that are monitored by the Brazilian Unified Health System (SUS). This study was conducted within the scope of the RRRL/MoH as part of a Federal public health policy for viral AGE surveillance in Brazil. Patient-informed consent was waived by the Fiocruz Ethical Committee, and patients’ data were maintained anonymously and securely.

### 2.3. Nucleic Acid Extraction

Viral nucleic acids (DNA and RNA viruses) were purified from 140 µL of clarified stool suspension (10% *w*/*v*) prepared with Tris-calcium buffer (pH = 7.2). Samples were subjected to an automatic nucleic acid extraction procedure using a QIAamp^®^ Viral RNA Mini kit and a QIAcube^®^ automated system (both from QIAGEN, Hilden, Germany) according to the manufacturer’s instructions. Viral nucleic acid was eluted in 60 µL of the elution buffer AVE. The isolated RNA was immediately stored at −80 °C until the molecular analysis. In each extraction procedure, RNAse/DNAse-free water was used as a negative control.

### 2.4. RVA Detection, Genotyping, and Sequencing

RVA was detected and quantified using a TaqMan^®^-based quantitative one-step PCR (RT-qPCR) with primers and a probe targeting the conserved NSP3 segment, according to Zeng et al. [21]. RVA-positive samples obtained using RT-qPCR were G and P genotyped using a one-step multiplex RT-PCR. The reactions were performed using the Qiagen One-Step RT-PCR kit (QIAGEN) and forward conserved primers VP7uF or VP4uF, as well as specific reverse primers for G types G1, G2, G4, G6, G3, G9, and G12 or P types P[4], P[6], P[8], P[9], and P[10] as recommended by the Centers for Disease Control and Prevention, USA. The G and P genotypes were assigned based on different amplicon sizes and base pairs (bp) using agarose gel analysis [22,23,24]. For the G6 genotype detection, the forward primer G6a [25] together with the reverse primer VP7-R [26] were used, generating an amplicon size of 196 bp.

Sanger sequencing was also used to characterize the nucleotide (nt) sequence of specific RVA strains. VP7 and VP4 genes were amplified using consensus primers of Beng9/End9 and 4Con3/4Con2, generating amplicons of 1062 bp and 876 bp, respectively [22,27]. The generated amplicons were purified using the ExoSAP clean-up kit (Thermo Fisher Scientific, Waltham, MA, USA) or the QIAquick Gel Extraction Kit (QI-AGEN), and purified amplicons were sequenced in both directions at the FIOCRUZ Institutional Platform for DNA sequencing (PDTIS).

### 2.5. Phylogenetic Analysis

Chromatogram analysis and consensus sequences were obtained using the Geneious Prime software (Biomatters Ltd., Auckland, New Zealand). The genotypes of VP7 and VP4 were identified using the Rotavirus A Determination tool available in Virus Pathogen Resource (ViPR; https://legacy.viprbrc.org/brc/home.spg?decorator=vipr, accessed on 15 February 2023) [28]. The Basic Local Alignment Search Tool (BLAST) server [29] was also used as a complementary tool for genotype identification. Multiple alignments of the sequences were performed using the CLUSTAL W program in MEGA 11 v. 11.08 [30].

Phylogenetic trees of the VP7 and VP4 genes were constructed using the maximum likelihood method and the selected best-fit evolutionary model for the data set via the Kimura two-parameter model (2000 bootstrap replications for branch support) in MEGA 11 v11.08, using RVA reference sequences obtained from the National Center for Biotechnology Information (NCBI) GenBank database. Nucleotide sequences obtained in this study were deposited in the GenBank database under the accession numbers OR233212-OR233269 and OR187100-OR187151.

### 2.6. Statistical Analysis

Statistical analyses were performed using GraphPad Prism software v9.5.1 (GraphPad Software, San Diego, CA, USA). As appropriate, the Mann–Whitney U test, chi-squared test, or Fisher’s test were used to assess significant differences between RVA detection rates, years of collecting samples, and age groups, as well as to compare RVA viral load according to different age groups. A *p*-value < 0.05 was considered to be statistically significant.

## 3. Results

### 3.1. Rotavirus A Epidemiology

Between January 2020 and December 2022, we tested RVA in a total of 2173 stool samples, received from medically attended AGE cases. Overall, RVA was detected in 7.7% of samples (n = 167): 15.5% (47/303) in 2020, 0.5% (5/1035) in 2021, and 13.8% (115/835) in 2022. RVA was detected in 57.3% and 44.2% of samples from males and females, respectively (*p* = 0.548). Monthly detection rates varied from 0.5% in October 2021 to 48.1% in September 2020, with the latter being the highest detection rate observed during the period. Regarding annual circulation, we did not observe a marked seasonality. However, during the triennium, three RVA circulation peaks were observed in the months of September 2020/2022 and November 2022 (Figure 1).

Concerning regional analysis, a higher RVA prevalence was observed in the Northeastern region (19.3%) compared to the Southeastern and Southern regions (6.1% and 5.5%, respectively). RVA detection rates were higher in 2020 and 2022 in the Northeastern, Southeastern, and Southern regions compared with 2021 in all regions. Additionally, during the three-year study period, the two states of the Southern region (Santa Catarina and Rio Grande do Sul) sent the largest number of samples for analysis and represented almost half the number of RVA-positive samples, 46.7% (78/167). Table 1 shows a detailed analysis of RVA detection by regions and states.

The majority of RVA-positive samples were from children less than five years old, representing 64.1% (107/167). The most affected age group was children aged between 0 and 6 months old; however, the score was not statistically significant (*p* = 0.3505). The other detection rates varied from 6.6% in the group of children from >12 to 24 m old to 8.5% in children >6 to 12 m old. Regarding patients within the >60 m group, comprised of older children, adolescents, and adults and elders (≥65 y), RVA was detected in 2.3% (20/846), 3.8% (32/846), and 0.9% (8/846) of samples, respectively (Table 1).

### 3.2. RVA Genotyping

We successfully genotyped (G and P genes) 98.2% (164/167) of positive samples using multiplex RT-PCR. Of these, 45 samples were from 2020, 5 were from 2021, and 114 were from 2022. We identified seven different RVA genotypes circulating during the period: G1P[8], G3P[8], G3P[9], G6P[8], G9P[4], G9P[8], and G12P[6]. G3P[8] was the most prevalent genotype during the first year of the study, being detected in 72.3% (n = 34) of RVA-genotyped samples; this was replaced by G6P[8] in 2022 after it was detected in 83.5% (n = 96) of genotyped samples. Usual RVA genotypes G1P[8] and G9P[8] were detected in one sample in 2022 and in two samples in 2020, respectively (Figure 2).

We also detected unusual G/P combinations, such as G3P[9] in one sample from 2021, G9P[4] in two samples from 2022, and G12P[6] in two samples from 2020. G and/or P non-typed (NT) accounted for 4.2% of samples and were represented mostly by samples with a low viral concentration (usually Ct values above 30) (Figure 2). RVA genotype distribution did not differ significantly by age group (Figure 3).

### 3.3. VP7 and VP4 Phylogenetic Analysis

In addition to genotyping, we sequenced selected samples in order to gain more information on the circulating strains and their respective lineages. We successfully obtained 59 and 52 consensus sequences of VP7 and VP4 genes, respectively. We conducted a phylogenetic analysis based on the nucleotide sequences of the near-complete ORFs of VP4 and VP7 genes in order to determine the genetic relationship between Brazilian strains and other RVA strains from GenBank. Initially, we confirmed the genotypes obtained using multiplex RT-PCR on Brazilian strains belonging to G1P[8], G3P[8], G3P[9], G6P[8], G9P[4], G9P[8], and G12P[6] and, through phylogenetic analysis, determined their respective lineages.

For the G1 genotype, the Brazilian G1P[8] sequence clustered within lineage II and showed the highest nucleotide similarity (100%) with the vaccine-derived rotavirus G1P[8] strain from Japan (KY616899: RVA/Human-wt/JP/JP11786/2013/G1[8]), as well as with the Rotarix™ vaccine strain (RVA/Human-tc/USA/Rotarix/1998/G1P1[8]; H6917354). The G3 Brazilian sequences clustered into lineage IX, comprising equine-like G3P[8] sequences. Brazilian G3 sequences were genetically related to equine-like G3P[8] strains from Brazil (RVA/Human-wt/BRA/LVCA30171/2019/G3P[8]; MT063065) (RVA/Human-wt/BRA/AM16-31/2016/G3P[8]; KX469400) and other countries, such as Germany (RVA/Human-wt/GER/GER 37-16/2016/G3P[8]; KX000546) and Thailand (RVA/Human-wt/THA/MS2014/2014/G3P[8]; LC455770); moreover, nucleotide similarities varied from 99.2% to 99.4%. One Brazilian G3 sequence clustered into lineage I, which comprised Wa-like G3P[8] sequences. This Brazilian sequence showed the highest nucleotide similarities (98.6–99.7%) with G3 sequences from Russia (RVA/Human-wt/RUS/Nov06-k2/2006/G3P[9]; FJ4352206) and Italy (RVA/Human-wt/ITA/PA307/2011/G3P[9]; MW280955) (Figure 4).

The emergent G6 sequences detected in our study, harboring a P[8]-type, clustered into lineage I and showed the highest nucleotide similarities (ranging from 97.5 to 99.1%) with a Brazilian G6P[8] strain detected in 2019 (RVA/Human-wt/BRA/LVCA30319/2019/G6P[8]; OP004924), as well as sequences from Bulgaria (RVA/Human-wt/BG/BG105/2010/G6P[8]; KM590373), Germany (RVA/Human-wt/GER/GER29-14/2014/G6P[9]; KX880436), and Belgium (RVA/Human-wt/BEL/B1711/2002/G6P[9]; EF554087) (Figure 4). The Brazilian G9 strains clustered within two G9 sequences from the USA (RVA/Human-wt/USA/R160/1999/G9; AF274971) (RVA/Human-wt/USA/ R136/2021/G9; AF438228). Other G9 sequences (lineages I and II) from different countries were segregated into two different clusters, with nucleotide similarities ranging from 89.3% to 90% (Figure 4). The one G12 sequence detected in our study clustered into lineage III and formed a monophyletic cluster with one Brazilian G12P[6] strain detected in 2019 (RVA/Human-wt/BRA/LVCA30622/2019/G12P[6]; OP004921). Our strain also clustered together with G12P[6] sequences from Thailand (RVA/Human-wt/THA/CMHN49-12/2012/G12P[6]; KT936631), Hungary (KT007579: RVA/Human-wt/HU/CU-B1373/2012/G12P[6]; KT007579), Germany (RVA/Human-wt/GER/GER172-08/2008/G12P[6]; FJ747630), and Bangladesh (RVA/Human-wt/BAD/SK277/2005/G12P[6]; EU839943) (Figure 4).

Regarding the VP4 gene, the two P[4] Brazilian strains clustered into lineage III and showed the highest nucleotide similarities (ranging from 99.5 to 99.6%) with P[4] sequences from Pakistan (RVA/Human-wt/PAK/PAK647/2016/G3P[4]; MH236897) and Czech Republic (RVA/Human wt/CZE/H186/2018/G9P[4]; MT005297), isolated in 2016 and 2018, respectively (Figure 5). For the P[6] genotype, our strains clustered into lineage I and showed the highest nucleotide similarities (>99.8%) with sequences from Belgium (RVA/Human-wt/BE/3001607814/2018/G12P[6]; MZ066324) and Brazil (RVA/Human-wt/BRA/LVCA30622/2019/G12P[6]; OP004927). Other P[6] sequences within lineage I were segregated into two different branches, showing nucleotide similarities ranging from 99 to 99.5% (Figure 5).

A total of 47 Brazilian P[8] sequences from this study clustered into lineage III; moreover, within this lineage, the highest nucleotide similarities (99.5–99.9%) were observed with a Brazilian strain that was isolated in 2015 (RVA/Human-wt/BRA/IAL-R330/2015/G3P[8]; MH569766), as well as strains from other countries such as Germany (RVA/Human-wt/GER/GER33-15/2015/G3P[8]; KX880415), Hungary (RVA/Human-wt/HUN/ERN8263/2015/G3P[8]; KU870410), Spain (RVA/Human-wt/ESP/SS98242319/2015/G3P[8]; KU550280), Taiwan (RVA/Human-wt/TWN/DO73/2016/G3P[8]; MF044167), and Dominican Republic (RVA/Human-wt/DOM/3000503701/2014/G3P[8]; MG652333) (Figure 5). The other Brazilian P[8] sequence clustered into lineage I, with the highest nucleotide similarity (>99.7%) found with a Japanese G8P[1] strain (RVA/Human-wt/JPN/OSN9-RX/2014/G1P[8]; LC028931) and the Rotarix™ vaccine strain (RVA/Vaccine/USA/Rotarix/1988/G1P1A[8]; JN849113) (Figure 5). Brazilian P[9] sequences, in combination with G3, clustered in lineage I and showed the highest nucleotide identity (98.9–99.9%) with G3P[9] Russian and South Korean strains [RVA/Human-wt/RUS/RoV 16477/2019/G3P[9]; (ON603988); RVA/Human-wt/RUS/NN1241-20/2020/G3P[9]; (KJ919655); RVA/Human-wt/KR/CAU-1202051/2014/G3P[9]; (KJ187602)], as well as with a Japanese G6P[9] strain (RVA/Human-wt/JPN/KF17/2010/G6P[9]; JF421978) (Figure 5).

## 4. Discussion

Our three-year study provides information on the prevalence and diversity of RVA genotypes detected from medically attended AGE patients (symptomatic children and adults) collected in eight states from three regions in Brazil during the COVID-19 pandemic. Overall, we detected RVA in 7.7% of samples. We also identified the emergence and predominance of the G6P[8] genotype, especially in the second and third years of the study.

Previous studies performed in the North and Midwest regions of Brazil demonstrated RVA detection rates varying from 24.2% to 33% during 2011 and 2015, and from 9.9% to 25.3% during 2013 and 2017 [31,32,33]. A previous study from our group reported an RVA prevalence of 20.8% among children aged up to 12 years, collected between 2006 and 2017, with annual detection rates ranging from 5% to 40% [34]. More recently, we demonstrated RVA detection rates of 10.5% and 13.5% in Brazil, with samples from 2018 and 2019, respectively [35]. The lower RVA detection rates observed in the present study compared to others in Brazil performed earlier likely reflect the substantial reduction in RVA notifications during the COVID-19 pandemic. Additionally, some of those studies involved only hospitalized children, who typically have higher rates of RVA infection than older children and adults and patients treated in outpatient settings. Similar to other countries, non-pharmaceutical interventions implemented in Brazil to control the spread of SARS-CoV-2, such as wearing masks, surface disinfection, social distancing, hand hygiene, and school closure, helped to contain other infectious agents, such as RVA [36]. In line with our results, other countries also reported a lower circulation of RVA during the COVID-19 pandemic compared to previous years, such as Australia [37], China [38,39], Finland [40], Japan [41], and the USA [42]. A study in Hangzhou, China, found that the detection rate of RVA among children with AGE symptoms decreased in 2020 compared to 2019 [39]. The RVA positivity rate in 2020 was 7.1%, which is significantly lower than the rate of 14.4% observed in 2019 [39]. Similarly, in Thailand, RVA was detected in 11.9% of samples from hospitalized children with AGE in 2019 and 2020 [43], which is similar to the RVA positivity rate observed in our study.

A study conducted in Japan, from July 2014 to June 2020, reported an RVA detection rate of 0% among children with AGE symptoms during the 2019–2020 period [41]. The authors noted that the number of stool samples received during that period was much smaller compared to other periods, likely due to the infection control measures against COVID-19 in Japan [41]. Similarly, another study in China reported a significant decline in the RVA positivity rate during the winter of 2019–2020, which remained at a much lower level of 0.1–0.6% in September 2020 compared to the RVA incidence observed in the same period over the previous seven years [38]. Regarding RVA seasonality, previous studies in Brazil have demonstrated that RVA circulates year-round, with peaks of detection in the winter/spring months (usually in September) [34,35,44]. In the present study, the RVA circulation pattern was likely affected by the lockdowns and restriction measures adopted during the early phase of the COVID-19 pandemic.

In the second year of the COVID-19 pandemic, there was an increasing demand to lift restrictions [45]. In many countries, the restrictions were relaxed and raised concerns about a potential rapid rise in infectious diseases, including RVA. Mathematical modeling studies suggest that, if contact patterns return to pre-pandemic levels, the population’s susceptibility to RVA and norovirus infections would likely increase [46,47]. During our study, the RVA detection rate decreased dramatically in 2021 (0.5%) to a rate that was significantly lower than the rates observed in previous years in Brazil [34,35]. Our results are in line with reports from Finland, Norway, and the USA [40,42,48], which showed that RVA detection decreased rapidly between 2020–2021, when COVID-19 mitigation measures were in place. In 2022, we detected an RVA positivity rate of 13.8%, similar to pre-pandemic levels in Brazil, and RVA detection rates of 10.5% and 13.7% in 2018 and 2019, respectively, were observed [35].

The Rotarix™ vaccine was introduced in Brazil in March 2006, resulting in a significant reduction in childhood hospitalization due to severe AGE [49,50]. Despite a slight decline in vaccination coverage in Brazil from 2016 (89%) to 2017 (85.1%), there was an increase in coverage in 2018, reaching 91.3% [19,51]. However, since then, there has been a downward trend in RVA vaccine coverage. In 2019, the World Health Organization (WHO) identified “vaccine hesitancy” as one of the top 10 threats to global health (https://www.who.int/news-room/spotlight/ten-threats-to-global-health-in-2019, accessed on 28 June 2023). This was reflected in decreased vaccination rates, including for RVA vaccines [19]. During the COVID-19 pandemic, despite recommendations to maintain uninterrupted immunization services, there was a reduction in overall vaccine coverage worldwide [17,18]. In Brazil, according to preliminary data from the Information System of the National Immunization Program, RVA vaccination coverage was 73% in 2022, which was below the target rate of 90% (http://pni.datasus.gov.br/perguntas_si_pni.asp, accessed on 29 June 2023).

Regarding age groups, we found that the highest prevalence of RVA infections was observed in children aged 0 to 6 months compared to other age groups. The increased susceptibility of infants in this age group is likely attributed to the low RVA vaccination coverage during the COVID-19 pandemic. In our study, we observed that less than half of the RVA-positive cases among children under 5 years old were vaccinated (23%, n = 40) with one, two, or three vaccine doses.

Regarding RVA genotype characterization, we successfully identified G and P types in 98.2% of positive samples. G3P[8] strains predominated in 2020 (72.3%), being replaced by G6P[8] in 2022. Phylogenetic analysis of the VP7 and VP4 genes confirmed the characterization of G3 Brazilian strains belonging to lineage IX, as well as all the P[8] sequences to lineage III, which comprise equine-like G3 strains. After its first identification in Australia in 2013 [52], equine-like G3P[8] strains (possessing DS-1-like genetic constellations) spread and became endemic worldwide. Most of the recently emerged G3 rotaviruses in many countries possess DS-1-like RVA genome segments [33,53,54,55,56,57,58,59]. In Brazil, equine-like G3P[8] strains were the most prevalent and accounted for 83.8% in 2018 and 65.5% in 2019 among the genotyped samples [35]. Another study by our group identified G3P[8] strain genogroup 2 (G8-P[8]-I2-R2-C2-M2-A2-N1-T2-E2-H2) with an additional reassortment event in the NSP2 gene (N1) of Wa-like origin [60].

In our study, unusual G6P[8] strains emerged in 2020 and became the most predominant genotype in 2022. Phylogenetic analysis of the VP7 gene demonstrated that G6 Brazilian strains clustered into lineage I, which is associated with P[8]-III. Within lineage I, our strains clustered with the closest homology with strains from Belgium, Bulgaria, Germany, and Brazil [61,62,63,64]. The VP4 gene of P[8] Brazilian strains clustered into lineage III, with strains from Hungary, Germany, Spain, the Dominican Republic, and Brazil [33,53,54,55,65]. Previously, our group reported the detection of G6P[8] strains in 13.8% of AGE samples from 2019 [35]. The genotype G6 is occasionally detected in humans; however, it is commonly detected in bovines and is associated with a wide range of human and animal P genotypes [66,67]. P[8] and P[4] RVA genotypes have been identified as the first and second predominant P genotypes, respectively, causing more than 90% of RVA-associated AGE cases in many countries [68].

The RVA G6P[8] and G6P[4] genotypes have been detected in multiple countries, primarily focusing on the molecular characterization of the surface proteins (VP4 and VP7) [25,66,67,69,70,71]. In Brazil, RVA G6P[8] was identified in three children under the age of five who exhibited symptoms of AGE. Comprehensive genome characterization revealed that the G6P[8] strain exhibited a reassortant constellation of G6P[8]-I2-R2-C2-M2-A2-N2-T2-E2-H2 [64]. A study in Bangladesh identified a rare G6P[8] bovine-like RVA strain that was detected in a 7-month-old infant with AGE symptoms. Complete genotype analyses revealed that bovine G6 strains possessed the G6-P[8]-I2-R2-C2-M2-A11-N2-T6-E2-H3 genotype constellation [72].

We identified vaccine-derived G1P[8] RVA in one case of AGE in a 3-month-old infant in 2022, without information on the vaccination status. Recently, a study conducted in Brazil by Cunha et al. [73] demonstrated the shedding of the Rotarix™ G1P[8] virus among neonates and infants before receiving the first dose, indicating the transmission of vaccine-derived viruses from immunized to non-vaccinated children. Additionally, in the same study, the authors detected G1P[8] vaccine-derived RVA in one participant with AGE symptoms. In Australia, Roczo et al. [37] described the detection of vaccine-like RVA in 97.8% of the G1P[8] genotyped samples (n = 141) among infants aged between 0 and 6 months old.

In our study, we detected atypical genotypes G3P[9] and G9P[4] as minor genotypes. The G3P[9] combination is primarily associated with the AU-1 constellation and has been observed in reassortment events involving humans, felines, and cattle, highlighting its presence in both human and animal populations [63,74]. During a surveillance study, the RVA genotype G3P[9] was detected in two females aged 2 years old and 10 months old with AGE who had not received vaccination. The sample exhibited a genetic constellation of G3P[9]-I18-R3-C3-Mx-A19-N3-T3-E3-H6, indicating a reassortment event between humans and animals [75]. Another study conducted in Lebanon reported three rare human RVA strains, including two G3P[9] and one G3P[6], which were detected in stool samples from children under 5 years of age who were hospitalized for AGE [76]. Characterization of the complete genomes revealed that the G3P[9] strains exhibited a mixed DS-1/Wa/AU-1-like origin, suggesting their potential evolution through multiple reassortment events involving feline, human, and bovine RVA strains [76]. The other atypical genotype detected in our study, G9P[4], emerged in Japan, possibly through an independent reassortment event between G9P[8] and G2P[4] strains [77]. In Italy, Ianiro et al. [78] detected RVA G9P[4] strains in three young children exhibiting AGE symptoms. Nucleotide sequencing revealed that the genomic constellation of the three strains was G9-P[4]-I2-R2-C2-M2- A2-N2-T2-E2-H2, highlighting a human origin for all the investigated gene segments [78]. A surveillance study conducted in Argentina, between 2017 and 2018, identified atypical G9P[4] and G8P[8] strains (bearing DS-1-like genetic backbones) at moderate rates among hospitalized children with AGE [79].

Our study has some limitations. Firstly, the Brazilian RVA surveillance program encountered logistical challenges in collecting and shipping stool samples during the COVID-19 pandemic. Starting from April 2020, there was a significant decrease in reported cases of AGE. This decline can be attributed to the strict lockdown measures implemented by federal and state governments to contain the spread of SARS-CoV-2. These measures included city lockdowns, the closure of schools and daycare centers, physical distancing, increased hygiene practices, and the use of face masks. Furthermore, the overwhelmed healthcare system in Brazil, primarily attributed to the high number of COVID-19 cases, made it less likely for parents to seek medical assistance and report mild cases of AGE in children. Consequently, this led to the underreporting and potentially biased surveillance of AGE cases. Secondly, variability in reporting and collecting AGE cases among different states has further contributed to surveillance biases. Additionally, we were unable to explore differences in vaccine coverage and RVA prevalence among states and regions since such granular data are not available. Lastly, important RVA genes, such as VP6 and NSP4, were not characterized, and the complete gene constellation was not determined. Future studies are planned to conduct a comprehensive characterization of the complete genomes of the emerging G6P[8] strains and the unusual genotypes identified (G3P[9] and G9P[4]).

## 5. Conclusions

During the COVID-19 pandemic period, from 2020 to 2022, we identified RVA in 7.7% of samples obtained from medically attended patients exhibiting AGE symptoms in Brazil. In 2022, we observed the return to pre-pandemic levels of RVA positivity rate, as well as the emergence of the genotype G6P[8], which replaced the previous dominant genotype equine-like G3P[8]. Genomic surveillance of RVA plays a crucial role in identifying the major G and P genotypes circulating in Brazil, as well as in continuously evaluating their impact on the vaccination program. Additionally, the reduced coverage of RVA vaccination observed during the pandemic likely resulted in an increased pool of susceptible children. Therefore, it is necessary to maintain nationwide surveillance to monitor future trends and changes in the epidemiology of RVA in Brazil.

## Figures and Tables

**Figure 1 viruses-15-01619-f001:**
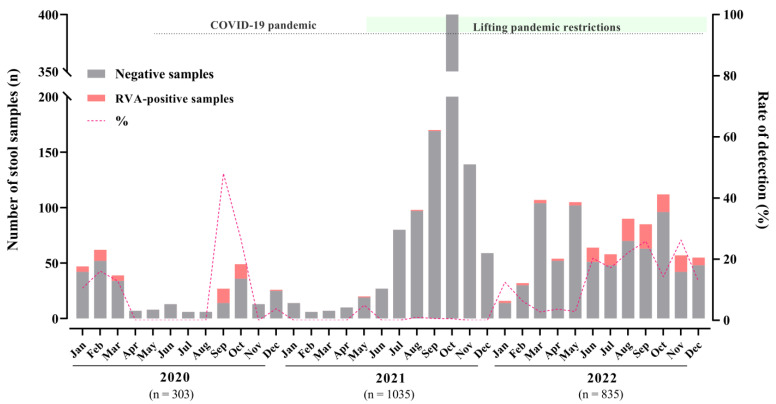
Monthly distribution of tested acute gastroenteritis samples, RVA-positive samples, and detection rates in Brazil during 2020–2022.

**Figure 2 viruses-15-01619-f002:**
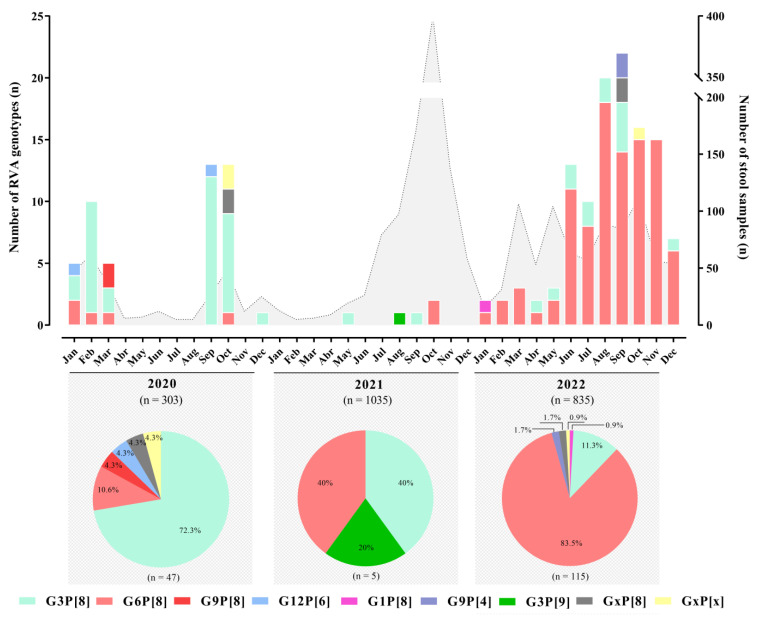
Distribution of RVA genotypes between January 2020 and December 2022. The monthly number of samples from children and adults diagnosed with AGE is shown in gray (right scale). Bar graphs show the number of RVA genotypes (left scale). Circle graphs show the percentage of RVA genotypes by year.

**Figure 3 viruses-15-01619-f003:**
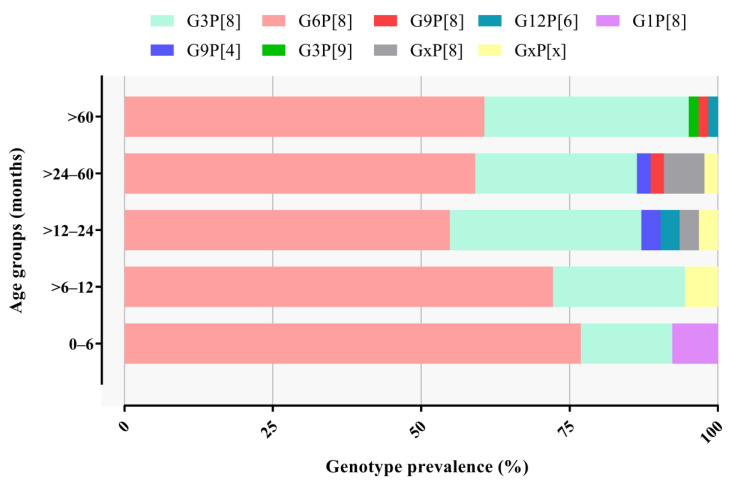
Distribution of RVA genotypes’ circulation (%) among different age groups of patients with AGE during the three-year study.

**Figure 4 viruses-15-01619-f004:**
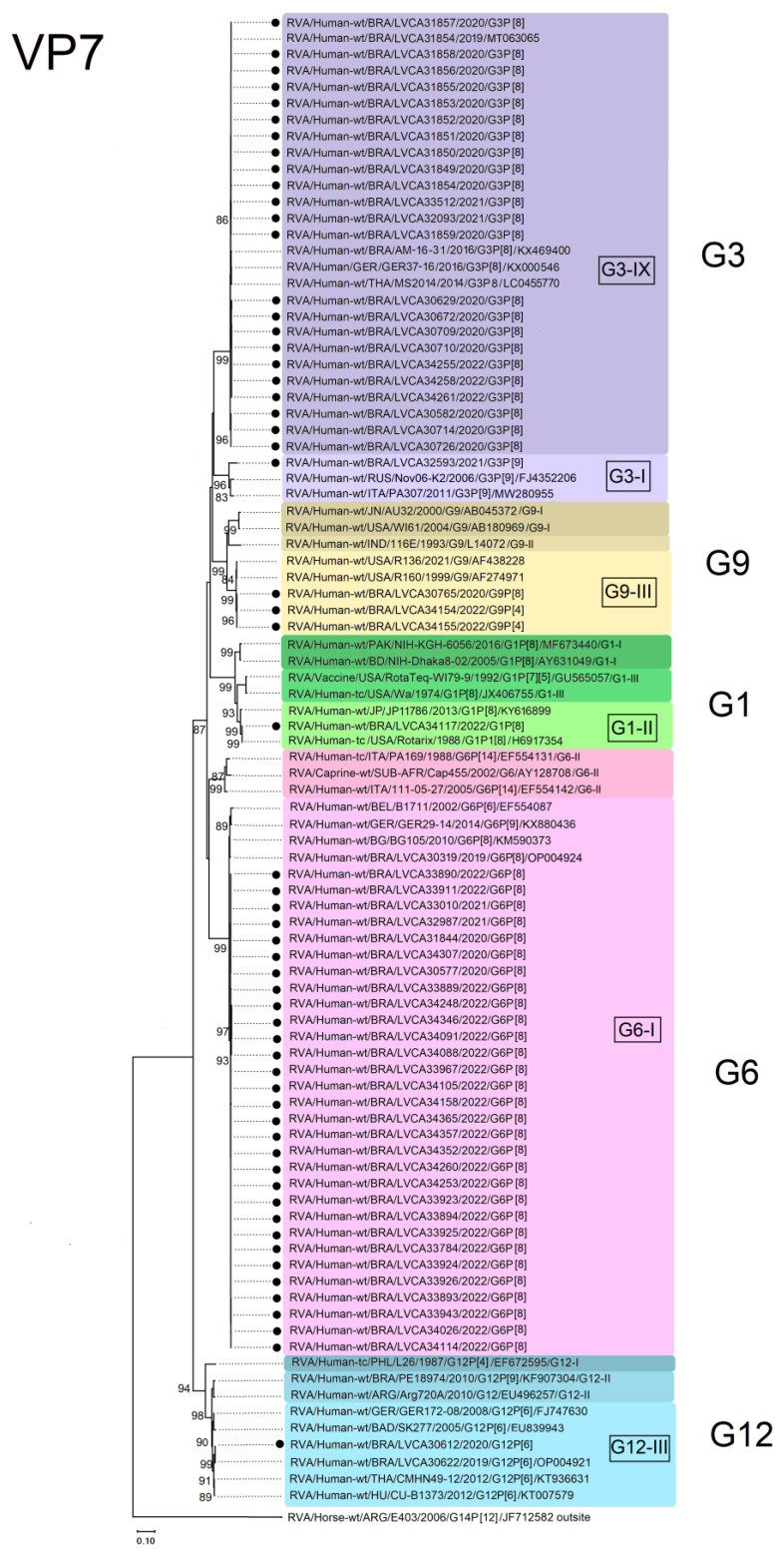
Maximum-likelihood phylogenetic tree method based on the Kimura 2-parameter model and the bootstrap tests (2000 replicates). The representative RVA strains are based on the VP7 gene fragment (n = 92). Brazilian strains obtained in this study are marked by black dots (n = 59). Bootstrap values above 85% are given at branch nodes. Distances in the tree are represented according to the scale at the bottom of the image.

**Figure 5 viruses-15-01619-f005:**
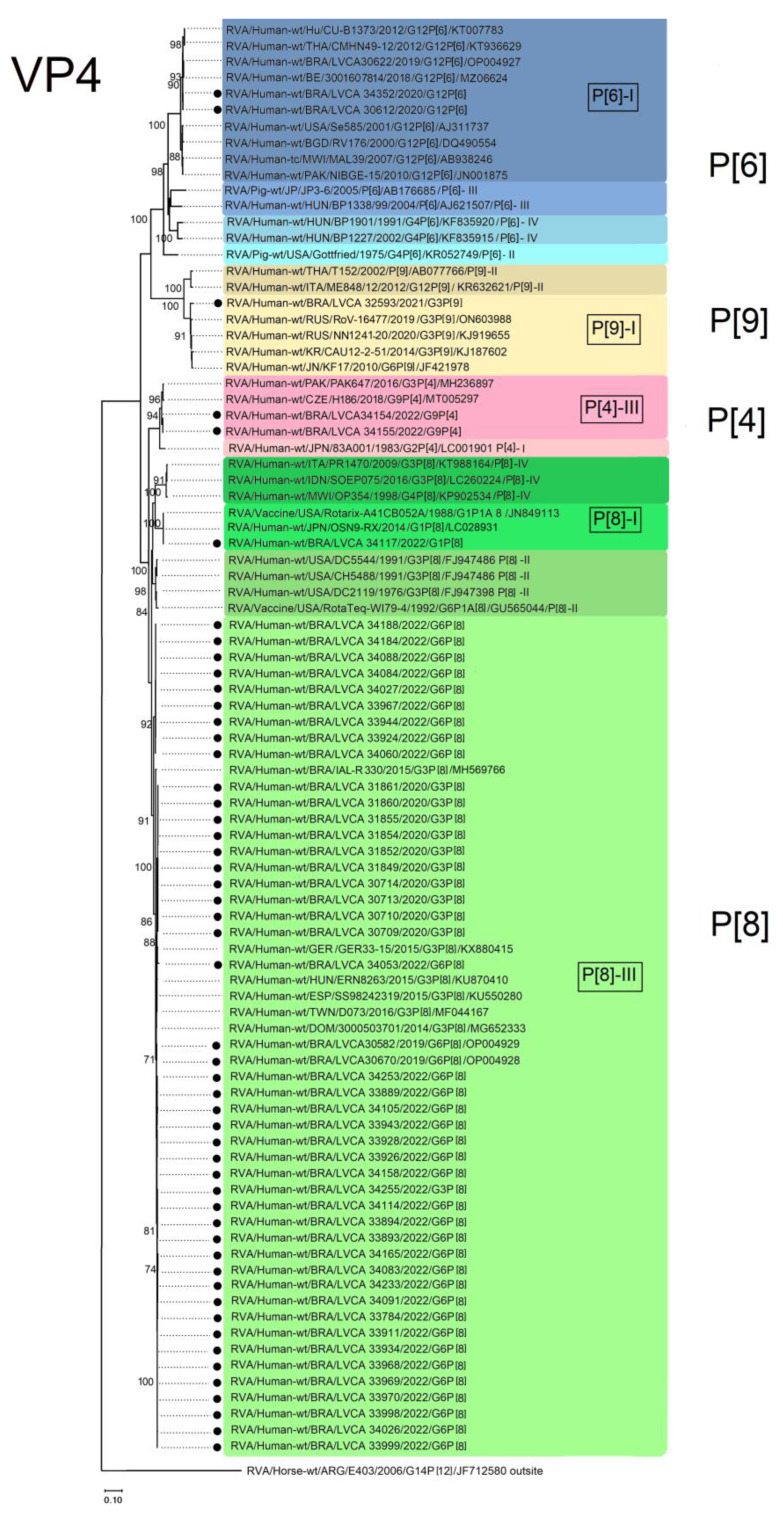
Maximum-likelihood phylogenetic tree method based on the Kimura 2-parameter model and the bootstrap tests (2000 replicates). The representative RVA strains are based on the VP4 gene fragment (n = 90). Brazilian strains obtained in this study are marked by black dots (n = 52). Bootstrap values above 85% are given at branch nodes. Distances in the tree are represented according to the scale at the bottom of the image.

**Table 1 viruses-15-01619-t001:** Number of rotavirus-positive samples through laboratory-based surveillance according to regions, states, and age groups in Brazil between 2020 and 2022.

	No. of Fecal Samples—Positive/Tested (%)				
	2020	2021	2022	Total	***p***-Value
Region/State					
Northeastern	25/106 (23.6)	3/108 (2.8)	34/108 (31.5)	62/322 (19.3)	<0.0001
Bahia (BA)	24/105	1/92	10/20		
Paraíba (PB)	0/0	0/0	4/6		
Pernambuco (PE)	1/1	2/15	19/81		
Sergipe (SE)	0/0	0/1	1/1		
Southeastern	9/61 (14.8)	1/142 (0.7)	17/240 (7.1)	27/443 (6.1)	0.1590
Minas Gerais (MG)	0/18	1/79	13/101		
Rio de Janeiro (RJ)	9/30	0/24	4/48		
Southern	13/136 (9.6)	1/785 (0.1)	64/487(13.1)	78/1408 (5.5)	<0.0001
Rio Grande do Sul (RS)	4/94	1/598	34/312		
Santa Catarina (SC)	9/42	0/187	30/175		
Age groups (months)					
0 to 6	3/33 (9.1)	0/52	10/48 (20.8)	13/133 (9.8)	-
>6 to 12	4/36 (11.1)	0/102	15/85 (17.6)	19/223 (8.5)	0.6212
>12 to 24	11/52 (21.2)	1/246 (0.4)	19/170 (11.2)	31/468 (6.6)	0.3305
>24 to 60	10/56 (17.9)	1/242 (0.4)	33/205 (16.1)	44/503 (8.7)	0.3076
>60	19/126 (15.1)	3/393 (0.8)	38/327(11.6)	60/846 (7.1)	0.4073

## Data Availability

The datasets generated and analyzed during the current study are available in the GenBank repository under accession numbers OR233212-OR233269 and OR187100-OR187151. This study is registered in the Brazilian National System for Genetic Heritage and Associated Traditional Knowledge Management (SisGen, No. A837EB6).

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
