# Peer review of "Rotavirus A during the COVID-19 Pandemic in Brazil, 2020–2022: Emergence of G6P[8] Genotype"

_viruses, 2023, doi:10.3390/v15081619_

Round 1

Reviewer 1 Report

This manuscript describes the detection of rotavirus and the circulating genotypes during the three-year COVID-19 pandemic from 2020-2022.  The data are clearly described and the limitations are appropriately noted.  

Specific comments:

-- Results, Lines 155-158:  Marked seasonality was not observed during these three years.  What is rotavirus seasonality in Brazil and how did it coincide with the implementation of the COVID-19 lockdowns and other prevention measures?  How does this compare to pre-vaccine seasonality.  Consider placing the seasonality findings in context in the discussion section.

--Figure 1:  There were no patterns in all-cause AGE seasonality observed and number of stools tested varied substantially over the 3 years.  Is the enrollment pattern more reflective of stool collection efforts?  What caused the sharp peak in stools tested in October 2021?

--Figure 1:  Is it possible to overlay COVID-19 pandemic control efforts (e.g. lockdowns, school closures, etc.) on Figure 1?

--Table 1:  The age group of enrolled cases is different from what is seen for most surveillance systems for rotavirus among children.  In such studies, rotavirus and all-cause AGE burden tend to skew to younger ages (6-18 months of age) with very few rotavirus and all-cause AGE cases are seen in children 2-5 years of age.  Is the age distribution seen in the current study a reflection of the types of hospitals that are collected specimens (e.g. adult hospitals with a children’s ward vs. children’s hospitals)?

--Discussion – lines 285-290:  The discussion starts with information that is better placed in the background section.  Start the discussion with the key findings from your study.

--Discussion – line 296-302:  It is very helpful to place the findings from your study in the context of other studies done in Brazil prior to the COVID-19 pandemic.  However, it is important to make sure that these studies are comparable.  The age of the population studied makes a huge difference in the prevalence of rotavirus.  Do the previous studies that you use for pre-pandemic comparators have a similar age distribution of enrolled cases?

There are a few grammatical issues and some orphan phrases (e.g. lines 177-178) throughout that could benefit from careful proofreading.

Author Response

Please, see attached. 

Reviewer 2 Report

In this study, Meylin Bautista Gutierrez et al. analyzed the diversity and prevalence of RVA genotypes in children and adults presenting with symptoms of acute gastroenteritis in Brazil during the COVID-19 pandemic in Brazil from 2020 to 2022. The results found that the prevalence of RVA was effectively controlled during the epidemic due to stricter national health policy requirements. At the same time, the emergence of the G6P [8] genotype replaced the previous dominant genotype equine-like G3P [8]. The results of this study are valuable for predicting future trends in RVA epidemiology in Brazil and provide a basis for research on high coverage and sustainable use of RVA vaccines. However, there are some limitations in this paper as follows.

1.       In the introduction, the necessity of conducting this study is not sufficiently described.

2.       SARA-CoV-2 infection is known to cause diarrheal symptoms similar to those of RVA infection in patients. To avoid data bias, please exclude the population with background of SARS-CoV-2 infection before performing the analysis.

3.       In Table 1, the prevalence of RVA was higher in the northeastern compared to the southeastern and southern regions. please provide an explanation.

4.       In the methods (page 3, lines 107-124), the authors used RT-qPCR and sanger sequencing for genotyping; is there consistency in the results of these two assays? Furthermore, Jelle Matthijnssens et al. (DOI 10.1007/s00705-008-0155-1) recommended that one of the criteria for RVA genotyping is "At least 50% of the ORF sequence should be determined", whereas the fragment sequenced for VP4 in this article is only 876bp, please explain this.

5.       The team had pointed out in their previous article (https://doi.org/10.3390/pathogens9070515) that "RVA detection rate was significantly higher among children aged between 24 and 60 months (18.8%) compared to the other age groups", please explain the reason for the non-difference in age prevalence trends in this article.

6.       In the discussion (page 12, lines 329-331), the statement " Our results are in line with the reports from Finland, Norway and USA, which showed that RVA detection decreased rapidly in 2020-2021, with COVID-19 mitigation measures". However, the epidemic control policy in Brazil has been reduced in 2021, please explain this.

In this study, Meylin Bautista Gutierrez et al. analyzed the diversity and prevalence of RVA genotypes in children and adults presenting with symptoms of acute gastroenteritis in Brazil during the COVID-19 pandemic in Brazil from 2020 to 2022. The results found that the prevalence of RVA was effectively controlled during the epidemic due to stricter national health policy requirements. At the same time, the emergence of the G6P [8] genotype replaced the previous dominant genotype equine-like G3P [8]. The results of this study are valuable for predicting future trends in RVA epidemiology in Brazil and provide a basis for research on high coverage and sustainable use of RVA vaccines. However, there are some limitations in this paper as follows.

1.       In the introduction, the necessity of conducting this study is not sufficiently described.

2.       SARA-CoV-2 infection is known to cause diarrheal symptoms similar to those of RVA infection in patients. To avoid data bias, please exclude the population with background of SARS-CoV-2 infection before performing the analysis.

3.       In Table 1, the prevalence of RVA was higher in the northeastern compared to the southeastern and southern regions. please provide an explanation.

4.       In the methods (page 3, lines 107-124), the authors used RT-qPCR and sanger sequencing for genotyping; is there consistency in the results of these two assays? Furthermore, Jelle Matthijnssens et al. (DOI 10.1007/s00705-008-0155-1) recommended that one of the criteria for RVA genotyping is "At least 50% of the ORF sequence should be determined", whereas the fragment sequenced for VP4 in this article is only 876bp, please explain this.

5.       The team had pointed out in their previous article (https://doi.org/10.3390/pathogens9070515) that "RVA detection rate was significantly higher among children aged between 24 and 60 months (18.8%) compared to the other age groups", please explain the reason for the non-difference in age prevalence trends in this article.

6.       In the discussion (page 12, lines 329-331), the statement " Our results are in line with the reports from Finland, Norway and USA, which showed that RVA detection decreased rapidly in 2020-2021, with COVID-19 mitigation measures". However, the epidemic control policy in Brazil has been reduced in 2021, please explain this.

Author Response

Please, see attached. 

Round 2

Reviewer 1 Report

The authors have adequately addressed my initial comments.  I only have a few small, minor comments on the revised manuscript.

--Thanks for adding the caveat to the discussion that may of the previous studies included only hospitalized children (lines 305-306).  However, it would be important to also include why this matters (e.g. Additionally, some of those studies involved only hospitalized children, who have typically have higher rates of rotavirus infection than older children and adults and patients treated in outpatient settings).

--There are a few grammatical errors that still need to be corrected (line 168 (A Concerning?), line 330 (there *was* an increasing demand, line 380 (…but *is* commonly detected), etc).

There are a few typos and grammatical errors throughout.

Author Response

The authors would like to thank the reviewer for the input during the second round of revision.

All the recommendations were implemented, and we thoroughly reviewed the manuscript to enhance the English language usage and rectify any remaining typos and grammatical errors.

Reviewer 2 Report

In this study, Meylin Bautista Gutierrez et al. analyzed the diversity and prevalence of RVA genotypes in children and adults presenting with symptoms of acute gastroenteritis in Brazil during the COVID-19 pandemic in Brazil from 2020 to 2022. The results found that the prevalence of RVA was effectively controlled during the epidemic due to stricter national health policy requirements. At the same time, the emergence of the G6P [8] genotype replaced the previous dominant genotype equine-like G3P [8]. The results of this study are valuable for predicting future trends in RVA epidemiology in Brazil and provide a basis for research on high coverage and sustainable use of RVA vaccines. However, there are some limitations, such as the lack of samples with a history of COVID-19 infection and a relatively small sample size. Overall, the article has a clear overall structure and elucidates the trend of RVA genotype variation in Brazil during the pandemic, which holds certain value.

In this study, Meylin Bautista Gutierrez et al. analyzed the diversity and prevalence of RVA genotypes in children and adults presenting with symptoms of acute gastroenteritis in Brazil during the COVID-19 pandemic in Brazil from 2020 to 2022. The results found that the prevalence of RVA was effectively controlled during the epidemic due to stricter national health policy requirements. At the same time, the emergence of the G6P [8] genotype replaced the previous dominant genotype equine-like G3P [8]. The results of this study are valuable for predicting future trends in RVA epidemiology in Brazil and provide a basis for research on high coverage and sustainable use of RVA vaccines. However, there are some limitations, such as the lack of samples with a history of COVID-19 infection and a relatively small sample size. Overall, the article has a clear overall structure and elucidates the trend of RVA genotype variation in Brazil during the pandemic, which holds certain value.

Author Response

The authors would like to thank the reviewer for the second round of revision, highlighting the value of our study.